# TGF-β/Smad Signalling Activation by HTRA1 Regulates the Function of Human Lens Epithelial Cells and Its Mechanism in Posterior Subcapsular Congenital Cataract

**DOI:** 10.3390/ijms232214431

**Published:** 2022-11-20

**Authors:** Xiaolei Lin, Tianke Yang, Xin Liu, Fan Fan, Xiyue Zhou, Hongzhe Li, Yi Luo

**Affiliations:** 1Department of Ophthalmology, Shanghai Eye Disease Prevention and Treatment Center, Shanghai Eye Hospital, Shanghai 200040, China; linxiaolei815@outlook.com; 2Department of Ophthalmology, Eye & ENT Hospital, Shanghai Medical College, Fudan University, Shanghai 200031, China; 20111260011@fudan.edu.cn (T.Y.); xin.liu@fdeent.org (X.L.); fanfan432121@hotmail.com (F.F.); 13301050257@fudan.edu.cn (X.Z.); 14301050163@fudan.edu.cn (H.L.)

**Keywords:** posterior subcapsular congenital cataract, HTRA1, TGF-β, lens epithelial cells

## Abstract

Congenital cataract is the leading cause of blindness among children worldwide. Patients with posterior subcapsular congenital cataract (PSC) in the central visual axis can result in worsening vision and stimulus deprivation amblyopia. However, the pathogenesis of PSC remains unclear. This study aims to explore the functional regulation and mechanism of HTRA1 in human lens epithelial cells (HLECs). HTRA1 was significantly downregulated in the lens capsules of children with PSC compared to normal controls. HTRA1 is a suppression factor of transforming growth factor-β (TGF-β) signalling pathway, which plays a key role in cataract formation. The results showed that the TGF-β/Smad signalling pathway was activated in the lens tissue of PSC. The effect of HTRA1 on cell proliferation, migration and apoptosis was measured in HLECs. In primary HLECs, the downregulation of HTRA1 can promote the proliferation and migration of HLECs by activating the TGF-β/Smad signalling pathway and can significantly upregulate the TGF-β/Smad downstream target genes FN1 and α-SMA. HTRA1 was also knocked out in the eyes of C57BL/6J mice via adeno-associated virus-mediated RNA interference. The results showed that HTRA1 knockout can significantly upregulate p-Smad2/3 and activate the TGF-β/Smad signalling pathway, resulting in abnormal proliferation and irregular arrangement of lens epithelial cells and leading to the occurrence of subcapsular cataract. To conclude, HTRA1 was significantly downregulated in children with PSC, and the downregulation of HTRA1 enhanced the proliferation and migration of HLECs by activating the TGF-β/Smad signalling pathway, which led to the occurrence of PSC.

## 1. Introduction

Congenital cataract is among the most common eye diseases causing blindness in children [1,2,3], accounting for 5–20% of cases [4,5,6]. Worldwide, about 200,000 children with blindness caused by congenital cataract, and approximately 20,000–40,000 children with congenital cataract are born every year [7,8,9]. Regional differences are significant in the incidence of congenital cataract, with the incidence in Asia being higher compared to other parts of the world at 7.43/1,000,000 [7]. Therefore, early prevention and treatment of congenital cataract are critical to reducing the burden on children, their families and society.

At present, surgery is the only effective way to treat congenital cataract. However, such surgery is difficult and expensive, and postoperative complications cannot be completely avoided. These complications can include visual axis opacification, secondary glaucoma, iris injury, and refractive error. Retinal detachment and endophthalmitis may even occur [10]. Therefore, it is crucial to clarify the pathogenesis of congenital cataract to prevent and predict the occurrence and development of this condition.

Different types of cataracts can lead to different degrees of visual impairment. Posterior subcapsular congenital cataract (PSC) is one of the most common types of congenital cataract. PSCs seriously affect visual development and account for 26.8% of congenital cataracts worldwide [7]. PSC is characterized by mass opacity under the posterior capsule without other ocular developmental abnormalities and is the simplest subtype of posterior polar cataract; however, the mechanism remains unknown. Therefore, the purpose of this study is to explore the potential mechanism of its occurrence and development. Previously, we performed transcriptome sequencing of capsules from patients with PSC and normal controls and found that HTRA1 was the most significant candidate pathogenic gene in lens epithelial cells [11]. HTRA1 has been previously detected in the lenses of mice [12]. Results based on the iSyTE2.0 database showed that the expression of HTRA1 in the lens of mice increased gradually with increasing gestational age, and the expression in the lenses was significantly higher than that in other tissues [13]. This indicates that HTRA1 may play an important role in lens development and cataract formation.

HTRA1 is a serine protease, and an important regulator of the TGF-β signalling pathway. GST pulldown experiments show that HTRA1 binds to many proteins of the TGF-β family and inhibits its function [14,15]. The relationship between HTRA1 and the TGF-β signalling pathway has been confirmed in a variety of diseases [16,17,18,19]. However, the function, expression characteristics and mechanism of HTRA1 in the lens have not been elucidated. Changes in the TGF-β signalling pathway represent an important pathogenesis in the formation of fibrous cataract [20]. Fibrous cataract mainly includes posterior cataract, anterior polar cataract and posterior polar cataract. The TGF-β superfamily members regulate important processes of lens development and promote the differentiation of lens fibre cells [21,22]. It has been considered that the TGF-β signalling pathway is key to the terminal differentiation of lens fibre cells [23]. TGF-β1 can induce rapid elongation of lens cells, abnormal accumulation of extracellular matrix, deposition of smooth muscle actin and apoptosis. TGF-β can affect the structure of the lens in the late stage of lens development, thus causing pathological changes in subcapsular opacification [24,25]. Transgenic mice with specific overexpression of lens TGF-β1 showed subcapsular plaque formation, posterior subcapsular nucleation and vacuole formation, which are similar to the pathological features of PSC [26,27,28]. However, whether changes in the TGF-β signalling pathway play a key role in the formation of PSC remains to be verified.

HTRA1 plays a key regulatory role in a variety of cell processes, such as cell proliferation, migration and apoptosis [29,30,31]. Previous nervous system studies have shown that the inhibition of HTRA1 protease activity activates the TGF-β signalling pathway, resulting in the upregulation of downstream target protein PAI-1 and eventually leading to neuronal apoptosis [16]. The expression of HTRA1 is downregulated in a variety of tumours, such as ovarian tumours [32], thyroid tumours [33] and non-small cell lung tumours [34], which activates the TGF-β signalling pathway to promote tumour cell proliferation [19]. However, the function of HTRA1 in lens epithelial cells needs to be further studied.

In the present study, we investigate the roles of HTRA1 and the TGF-β signalling pathway in the formation of PSC and lens epithelial cells.

## 2. Results

### 2.1. Downregulation of HTRA1 in Human Lens Epithelial Cells of Subjects with PSC

To verify the HTRA1 expression in human lens epithelial cells (HLECs), the anterior lens capsules of 36 eyes with PSC and 29 eyes with normal controls were analysed. Age (48.77 ± 29.35 months in the PSC group vs. 54.58 ± 36.65 months in the control group, *p* > 0.05) and sex (*p* > 0.05) were comparable for the two groups. Downregulation of HTRA1 was validated using human lens epithelial samples by RT-qPCR (*p* < 0.001, Figure 1A), Western blotting (*p* < 0.001, Figure 1B), and immunofluorescence staining (Figure 1C).

### 2.2. HTRA1 Expression Levels in the Normal Human Lens at Different Layers and in Different Age Groups

The expression levels of HTRA1 were measured in central lens epithelial cells, equatorial epithelial cells and fibre cells in normal controls. The level was highest in fibre cells and lowest in central lens epithelial cells (Figure 2A). Furthermore, HTRA1 decreased with age before 40 years of age in normal HLECs and was stable after 40 years of age (Figure 2B). The immunofluorescence staining of frozen sections showed that HTRA1 was expressed in both the cytoplasm and the nucleus of HLEC from normal controls (Figure 2C).

### 2.3. TGF-β/Smad Signalling Pathway Was Activated in HLECs of PSC Patients

To establish whether TGF-β/Smad signalling contributed to the formation of PSC, the RNA and protein expression of the key signalling molecules were determined. RT-qPCR results showed that TGF-β1 (*p* = 0.015) and TGF-β2 (*p* = 0.014) were upregulated in the lens epithelia of PSC patients. The key signalling molecules, including TGF-βR1 (*p* = 0.023), PAI-1 (*p* = 0.027) and CTGF (*p* = 0.002), were also significantly upregulated in PSC patients (Figure 3A). The protein levels of TGF-β2 (*p* = 0.008), TGF-βR1 (*p* = 0.040), p-Smad2/3 (*p* = 0.017) and PAI-1 (*p* = 0.040), the key signalling molecules, were significantly upregulated in lens epithelia of PSC patients (Figure 3B).

### 2.4. Downregulation of HTRA1 and Activated TGF-β/Smad Signalling Promotes HLEC Proliferation and Migration

HLEC were transfected with LV-HTRA1-RNAi lentivirus to downregulate the expression of HTRA1. After transfection, RT-qPCR and Western blotting were performed to verify the downregulation of HTRA1 (Figure 4A,B). A CCK-8 assay was used to explore the role of HTRA1 in the regulation of HLEC growth. The results showed that downregulated expression of HTRA1 significantly promotes the growth ability of HLEC (*p* < 0.001, Figure 5A). Moreover, a scratch healing assay showed that the knockdown of HTRA1 promoted the migration of HLEC (*p* = 0.022, Figure 5D). However, the ectopic expression of HTRA1 did not influence the apoptosis of HLEC (*p* > 0.05, Figure 5B,C).

Furthermore, HLECs were treated with 5 ng/mL TGF-β1 for 24 h. TGF-β1 treatment significantly promoted HLEC proliferation and migration, based on a CCK-8 assay and a scratch healing assay (Figure 6A,B).

In the mouse model, aberrant proliferation and migration of lens epithelial cells were also observed after knocking down HTRA1 by recombinant adeno-associated virus (rAAV) injection (Figure 7). Downregulation of HTRA1 induced the formation of subcapsular cataract (Figure 8).

### 2.5. Downregulation of HTRA1 Directly Activated TGF-β/Smad Signalling in Primary Lens Epithelial Cells

We investigated the possible regulatory relationship between HTRA1 and TGF-β/Smad signalling. In primary HLECs, the amount of colocalization between HTRA1 and TGF-β1, as well as HTRA1 and TGF-β2 was analysed after immunofluorescence staining. The Pearson’s correlation coefficient (PCC) was 0.62 for HTRA1 and TGF-β1 and 0.58 for HTRA1 and TGF-β2 (Figure 6E).

HTRA1 knockdown elevated the protein expression of p-Smad2/3, TGF-β1 and TGF-β2 in primary HLECs (*p* = 0.014, 0.021 and 0.039, respectively, Figure 4B,C). In contrast, the TGF-βR1/2 inhibitor attenuated the HTRA1-induced upregulation of p-Smad2/3 in primary HLECs (Figure 4D). And the growth and migration of HLEC were also attenuated by the TGF-βR1/2 inhibitor (Figure 5A,D), implying that the TGF-β/Smad pathway might play a role in regulating the aberrant proliferation and migration of HLECs induced by HTRA1. In addition, the downregulation of HTRA1 promoted the mRNA expression of the downstream target of TGF-β/Smad signalling, including α-SMA, PAI-1 and CTGF (*p* = 0.022, 0.005 and 0.038, respectively, Figure 4E). The protein expression of FN1 and α-SMA was also upregulated (*p* = 0.017 and 0.031, respectively, Figure 4F).

In primary mouse lens epithelial cells, HTRA1 knockdown also elevated the protein expression of p-Smad2/3, FN1 and α-SMA (*p* = 0.024, 0.002 and 0.002, respectively, Figure 9B,C).

## 3. Discussion

In this study, we found that HTRA1 is significantly downregulated in the lens capsules of PSC patients. In HLECs, the downregulation of HTRA1 expression can significantly promote cell proliferation and migration. The increase in the proliferation and migration of lens epithelial cells is closely related to the formation of cataract [21,35]. However, the mechanism of the effect of HTRA1 on cell proliferation and migration remains unclear. Wang et al. found that HTRA1 played an important role in regulating the stability and dynamics of microtubule assembly through the binding of the PDZ domain to microtubule [36]. It has also been found that the effect of HTRA1 on cell proliferation depends on the Notch-1 signalling pathway or the Wnt signalling pathway [37,38]. However, many studies have also suggested that the process of cell proliferation and migration regulated by HTRA1 is mediated by the TGF-β signalling pathway. HTRA1 can bind to multiple molecules of the TGF-β signalling pathway to inhibit its function, thus inhibiting cell proliferation, migration and invasion [19,39,40]. HTRA1 is closely related to members of the TGF-β family [39], and the TGF-β signalling pathway plays an important role in lens development and fibroblast differentiation [23]. 

The TGF-β signalling pathway was activated in the anterior capsule of PSC. In primary lens epithelial cells, downregulation of HTRA1 can promote the proliferation and migration of lens epithelial cells by activating the TGF-β/Smad signalling pathway and inducing significant activation of downstream FN1, α-SMA and other genes. Knockout of HTRA1 in the eyes of mice can activate the TGF-β signalling pathway and can subsequently cause subcapsular cataract. As a proteolytic enzyme, HTRA1 plays a role in protein quality control by identifying abnormal proteins and cutting them into smaller fragments [41]. HTRA1 plays a key role in many signalling pathways through the degradation of specific components. The results of in vitro experiments have shown that HTRA1 can bind to TGF-β family members and decompose TGF-β1, TGF-βR1, TGF-βR2, BMP2, BMP4 and other proteins [16,42]. When HTRA1 is downregulated, it causes the activation of TGF-β1 and its receptor, which leads to the significant upregulation of downstream p-Smad2/3. Many studies have obtained similar findings. For example, downregulation of HTRA1 leads to a significant increase in the expression of TGF-β1 during odontoblast differentiation, and the expression of downstream factors such as TGF-βR1, TGF-βR2, Smad2 and Smad4 are also significantly increased in the late stage of mineralization induction [43]. HTRA1 can inhibit bone formation by antagonizing the signal transduction mediated by TGF-β family proteins, regulating the activity of the TGF-β/BMP pathway and inhibiting the expression of downstream genes, such as RUNX2, CTGF and PAI-1 [39,42]. These results are consistent with the results of this study.

In the animal experiment, after HTRA1 knockdown we found subcapsular cataract and observed the abnormal proliferation of lens epithelial cells, with irregular cell arrangement and cell morphology. In addition, the fibre cells under the anterior capsule became loose and irregular. Robertson et al. injected an adenovirus vector carrying TGF-β1 into the anterior chamber of mice and found that adenovirus can successfully mediate the action of adenovirus on lens epithelial cells and cause abnormal proliferation of lens epithelial cells and cataract under the anterior capsule; this is consistent with the results of this study [44]. Compared with the adenovirus vector, ocular injection of the AAV vector can reduce the local and systemic immune response to the virus vector to the greatest extent and has stability in long-term expression [45]. The results of RT-PCR and Western blotting showed that AAV injection into the anterior chamber can successfully knock down the expression of HTRA1 in lens epithelial cells and activate the TGF-β signalling pathway. 

The results also showed that downregulation of HTRA1 can regulate the expression of downstream FN1, α-SMA, PAI-1 and CTGF. FN1, α-SMA, PAI-1 and CTGF are the markers of tissue fibrosis and epithelial–mesenchymal transition (EMT) [21,46,47,48], while EMT is the key factor in organ fibrosis [49]. TGF-β has been shown to mediate EMT in lens epithelial cells and to play a key role in lens fibrosis [50]. Anterior polar cataract is a kind of fibrous cataract. The expression of PAI-1 and α-SMA can be observed in the anterior capsule of anterior polar cataract [47]. In the process of EMT, the proliferation and migration of lens epithelial cells increase, which leads to the transformation of lens epithelial cells to fibroblasts, the deposition of α-SMA and the production of extracellular matrix, such as FN1, which eventually leads to the formation of plaques under the anterior capsule or posterior capsule of the lens [51,52]. In cell experiments, TGF-β can induce the expression of PAI-1 and mediate EMT induced by TGF-β1 [47,53]. At the same time, TGF-β2 can also induce lens epithelial cells to express α-SMA, type I collagen, type IV collagen and FN1 [48]. HTRA1 can degrade a variety of extracellular matrix components, such as type I, II and III procollagen peptides, fibronectin and proteoglycans, which play an important role in extracellular matrix remodelling [54,55]. Therefore, the downregulation of HTRA1 may mediate the expression of FN1, α-SMA and other proteins by activating the TGF-β/Smad pathway, resulting in the increased proliferation and migration of lens epithelial cells, lens fibrosis and PSC. However, the specific regulatory mechanism needs to be further studied.

There are some shortcomings in this study. One is that the specific regulation mechanism of HTRA1 needs to be explored more deeply. In addition, there is a lack of research on the ultrastructure of the lens in this animal experiment and on the effect of different injection sites on the morphology of cataract. The next step is to conduct an electron microscope examination on animal tissues and add the vitreous infection group to observe the changes in the morphology and the molecular level more comprehensively.

## 4. Materials and Methods

This study was approved by the Eye, Ear, Nose and Throat (ENT) Ethics Committee. Informed consent was obtained from each child’s guardian prior to surgery. All procedures followed the Declaration of Helsinki, and animal experiments were approved by the Ethics Committee for Animal Studies.

### 4.1. Patients and Tissue Sample Collection and Preservation

Patients under 10 diagnosed with bilateral or unilateral PSC and who were planning to undergo congenital cataract surgery at the Eye & ENT hospital, Shanghai, China were enrolled. The exclusion criteria were as follows: (1) ocular inflammation, (2) combined persistent foetal vasculature (PFV), persistent hyperplastic primary vitreous (PHPV) or pre-existing posterior capsule defect and (3) having a developmental disorder or a history of ocular surgery or trauma. The congenital cataract surgery was performed by the same surgeon (Y.L.). Anterior lens capsules of PSC patients were obtained using capsulorhexis during cataract surgery. The normal control group consisted of transparent lenses of children under 10 that were obtained from the Eye Bank. The anterior lens capsules were obtained using Vannas scissors and micro-forceps. To verify HTRA1 expression changes with age, normal adults without eye diseases were also included.

For RT-qPCR and Western blotting, samples were stored at −80 °C immediately after collection until use. For immunofluorescence staining, samples were collected in 4% paraformaldehyde (PFA). For primary culture, samples were kept in Dulbecco’s Modified Eagle Medium (DMEM, #12430054, Gibco, Grand Island, NY, USA) supplemented with 20% foetal bovine serum (FBS, #10099141, Gibco, Grand Island, NY, USA).

### 4.2. Construction of Recombinant Lentiviral Vectors and Adenoviral Vector

The GV493 vector (frame structure: hU6-CBh-gcGFP-IRES-puromycin, Genechem Co., Ltd., Shanghai, China) was used to construct the lentiviral vectors. Three recombinant lentiviral vectors expressing HTRA1-RNAi were constructed for the human HTRA1 gene knockdown: LV-HTRA1-RNAi (Lentivirus 1, 5′-GAAGTATATTGGTATCCGAAT-3′), LV-HTRA1-RNAi (Lentivirus 2, 5′-CGCCATCATCAACTATGGAAA-3′) and LV-HTRA1-RNAi (Lentivirus 3, 5′-CGTGGTTCATATCGAATTGTT-3′). A non-silencing targeting sequence (negative control: 5′-TTCTCCGAACGTGTCACGT-3′) was also cloned into GV493 as a control.

Recombinant adenoviral GV478 vectors (frame structure: U6-MCS-CAG-EGFP, Shanghai Genechem Co., Ltd., Shanghai, China) were constructed to silence the mouse HTRA1 gene. The target sequence is 5′-GGTTCACATTGAACTATATCG-3′, and the negative control sequence is 5′-CGCTGAGTACTTCGAAATGTC-3′. The working solution was prepared to 1 μL of the vehicle containing 2.0 × 10^10^ PFU.

### 4.3. Cell Line and Primary HLEC Culture

An HLEC line, SRA 01/04, was propagated in DMEM supplemented with 10% FBS and cultured at 37 °C in a 5 % CO_2_ atmosphere. The SRA 01/04 was obtained from Dr. Dan Li [56]. For the culture of primary paediatric HLECs, the lens capsules were attached to 35 mm cell culture dishes (Sigma-Aldrich Corp., Darmstadt, Germany) after collection during congenital cataract surgery, and 6 mm glass coverslips were used to cover the sample to promote attachment. About 0.5 mL DMEM supplemented with 20% FBS was carefully added to the surface of the samples. After 24 h, 0.5 mL culture medium was added, and the medium was changed every 3 d. The cultural atmosphere was the same as SRA 01/04. After reaching confluence, the cells were seeded on two or three culture dishes. SRA 01/04 was used for cell proliferation assay, scratch healing assay, as well as Annexin V/7-AAD staining and flow cytometry. And primary paediatric HLECs were used for the RNA and protein expression changes by RT-qPCR, Western blotting and immunofluorescence staining.

### 4.4. Lentivirus Transfection

A culture medium containing LV-HTRA1-RNAi was added to the cultured SRA 01/04 and primary HLEC. The cells were cultured for 72 h, and the expression of green fluorescent protein (GFP) was observed. The cells were then selected with 1.0 µg/mL puromycin after 48 h of incubation. The suppression of HTRA1 expression was analysed using RT-qPCR. The most efficient lentivirus was used for subsequent experiments.

### 4.5. Mouse Model and Sample Acquirement

Four-week-old male C57BL/6J mice obtained from JieSiJie Laboratory Animal Co., Ltd. (Shanghai, China) were used for in vivo gene transfer. DMEM containing 2 × 10^10^ rAAV was slowly delivered into the anterior chamber of the mice using a microinjector (Hamilton, Reno, NV, USA) with a 33-gauge needle (Hamilton, Reno, NV, USA). Animals were sacrificed and examined 4 weeks after rAAV administration. The lens capsules were obtained using Vannas scissors and micro-forceps.

### 4.6. RT-qPCR

Total RNA from the anterior lens capsules was extracted using TRIzol reagent (Invitrogen, Waltham, MA, USA). cDNA was then synthesized using the PrimeScript™ RT Kit and gDNA Eraser (TaKaRa, Tokyo, Japan). The mRNA expression levels were quantified by RT-qPCR (Bio-Rad, Hercules, CA, USA). The primers for RT-qPCR are listed in Appendix A. The 2^−ΔΔCt^ method was used to analyse the results.

### 4.7. Western Blotting and Immunofluorescence Staining

The protein levels of the samples were analysed using traditional Western blotting, the Wes-Protein simple system (ProteinSimple, Santa Clara, CA, USA) and immunofluorescence staining. The anterior lens capsules were lysed for total protein extraction by using radioimmunoprecipitation assay (RIPA) lysis buffer (Beyotime, Shanghai, China) supplemented with 1% cocktail protease inhibitors (MCE, Monmouth Junction, NJ, USA). The protein samples were denaturalized at 95 °C for 5 min, separated with sodium dodecylsulfate–polyacrylamide gel electrophoresis (SDS-PAGE) and transferred to a polyvinylidene difluoride (PVDF) membrane (Millipore, Burlington, MA, USA). The membrane was blocked with 5% non-fat milk and incubated with primary and secondary antibodies (Appendix A). It was then rinsed three times with TBST and visualized using an electrochemical luminescence kit (Thermo Fisher Scientific, Waltham, MA, USA). Band density was normalized to the loading control (β-actin) for further statistical analysis.

For immunofluorescence staining, anterior lens capsules or frozen sections were fixed in 4% PFA for 30 min. They were then permeabilized and blocked with PBS containing 0.3% Triton X-100 and 5% bovine serum albumin. Samples were then incubated with primary antibodies overnight at 4 °C. Subsequently, the secondary antibody and DAPI were applied. The samples were attached to the slides and observed using a confocal microscope (Leica, Richmond, IL, USA). The amount of colocalization between HTRA1 and TGF-β1, as well as HTRA1 and TGF-β2, was analysed using ImageJ processing software (v1.53c). The antibodies and dilution information are listed in Appendix A.

### 4.8. Cell Proliferation Assay

Cell proliferation was detected using Cell Counting Kit-8 (CCK8, Dojindo Laboratories, Kumamoto, Japan). After lentivirus transfection, HLECs (SRA 01/04) were grown in 96-well plates with 5 × 10^3^ cells per well. After incubation for 24 h, a CCK8 assay was performed at 37 °C. The optical density (OD) value at 450 nm was measured every 1 h and three times.

### 4.9. Scratch Healing Assay

After lentivirus transfection, HLECs (SRA 01/04) were seeded into 6-well plates and cultured until 90% confluent. The cells were then scratched carefully with 200 μL pipette tips at multiple sites. The suspended cells were washed with PBS, and the culture medium was changed to DMEM containing 1% FBS. The area of wound healing was observed and photographed at 0 h and 24 h after scratching at the same area.

### 4.10. Annexin V/7-AAD Staining and Flow Cytometry

After lentivirus transfection, HLECs (SRA 01/04) were digested into cell suspensions and adjusted to 1 × 10^6^~3 × 10^6^ cells/mL. The cells were washed with PBS, and 5 μL Annexin V-APC or 10 μL 7-AAD was added following the manufacturer’s protocol (Multisciences Biotech Co., Ltd., Taiwan, China). A flow cytometer (Becton, Dickinson and Company, Lake Franklin, NJ, USA) was used to test immediately after the reaction was completed.

### 4.11. TGF-β1 and TGF-βR1/2 Inhibitor Treatment

Primary HLECs were treated with recombinant human TGF-β1 (7666-MB-005, R&D Systems, Minneapolis, MN, USA) in a concentration gradient of 10 ng/mL for a duration of 24 h.

The selective TGF-βR1/2 inhibitors (LY2109761, Selleck, Shangai, China) were added to primary HLECs after lentivirus transfection at a working concentration of 10 μM for 24 h.

### 4.12. Haematoxylin and Eosin Staining

The eyeballs of mice were fixed in paraformaldehyde, and paraffin sections were made for further staining. The sections were stained with standard haematoxylin–eosin staining, and a microscope (Carl ZEISS, Oberkochen, Germany) was used to observe and capture images.

## 5. Conclusions

HTRA1 is significantly downregulated in lens epithelial cells of PSC. HTRA1 can promote the proliferation and migration of lens epithelial cells by activating the TGF-β/Smad signalling pathway, which may play an important role in the pathogenesis of PSC.

## Figures and Tables

**Figure 1 ijms-23-14431-f001:**
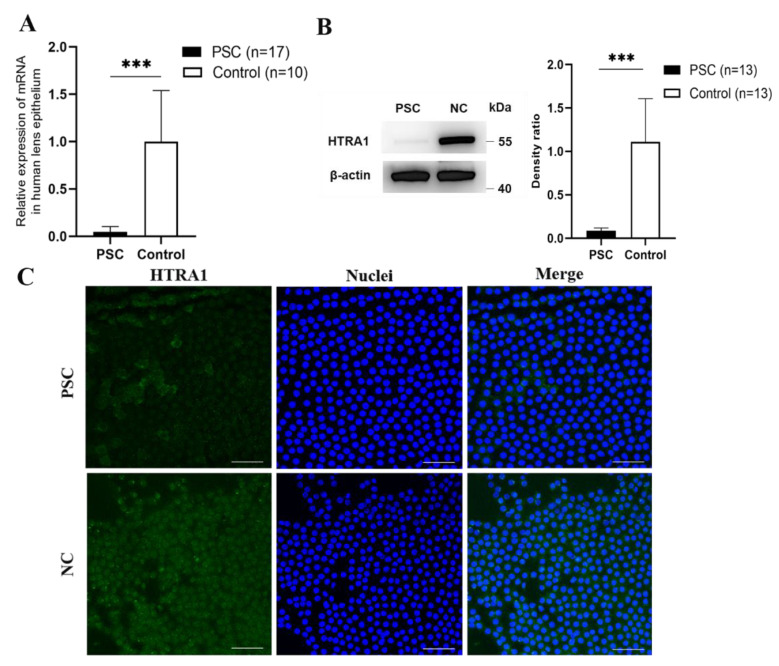
HTRA1 expression downregulated in PSC patients. (**A**) The mRNA level of HTRA1 in anterior lens capsules based on RT-qPCR analyses (*p* < 0.01). (**B**) The protein levels of HTRA1 in anterior lens capsules based on Western blotting (*p* < 0.001). (**C**) Immunofluorescence images of HTRA1 staining in anterior lens capsules (*n* = 3). *** *p* < 0.001, Scale bar: 20 μm.

**Figure 2 ijms-23-14431-f002:**
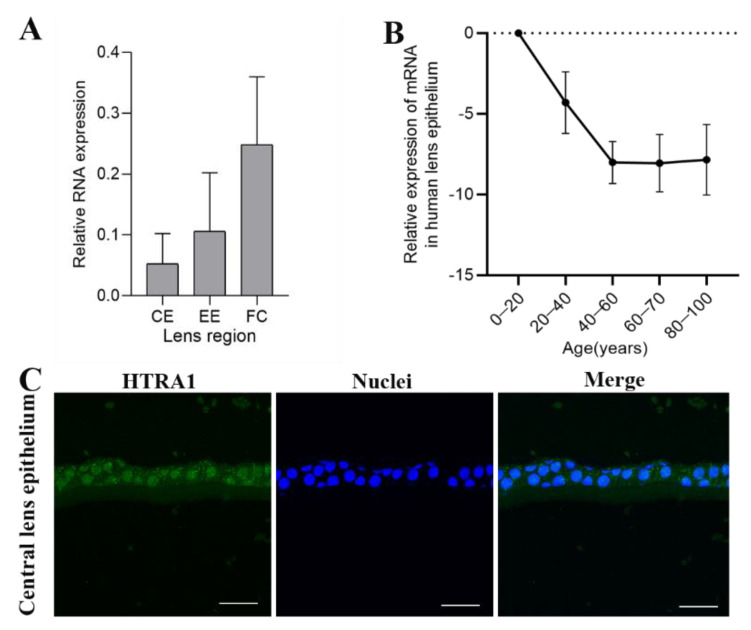
HTRA1 expression levels in normal human lenses at different layers and for different age groups. (**A**) The mRNA levels of HTRA1 expressed in central lens epithelial cells, equatorial epithelial cells and fibre cells in normal human lenses. (**B**) The mRNA levels of HTRA1 expressed in different age groups. (**C**) Immunofluorescence images of HTRA1 staining in HLECs (*n* = 3). Scale bar: 10 μm.

**Figure 3 ijms-23-14431-f003:**
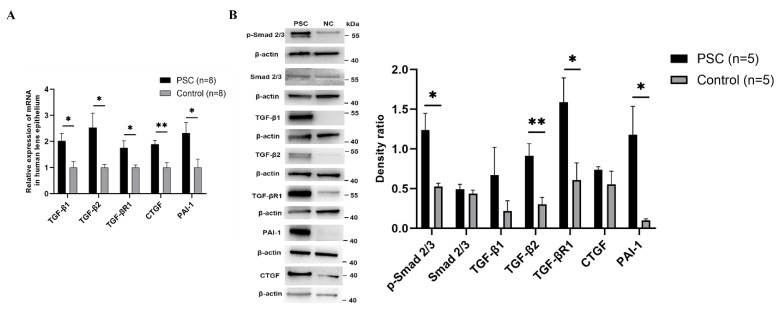
TGF-β/Smads signalling activated in lens of PSC patients. (**A**) The mRNA levels of TGF-β1, TGF-β2, TGF-βR1, CTGF and PAI-1 in human lens epithelial cells based on RT-qPCR analyses (*p* = 0.015, 0.014, 0.023, 0.002 and 0.027, respectively). (**B**) The protein expression of TGF-β1, TGF-β2, TGF-βR1, CTGF, PAI-1, Smad2/3 and p-Smad2/3 (*p* = 0.261, 0.008, 0.040, 0.321, 0.040, 0.487 and 0.017, respectively) based on Western blotting. * *p* < 0.05, ** *p* < 0.01.

**Figure 4 ijms-23-14431-f004:**
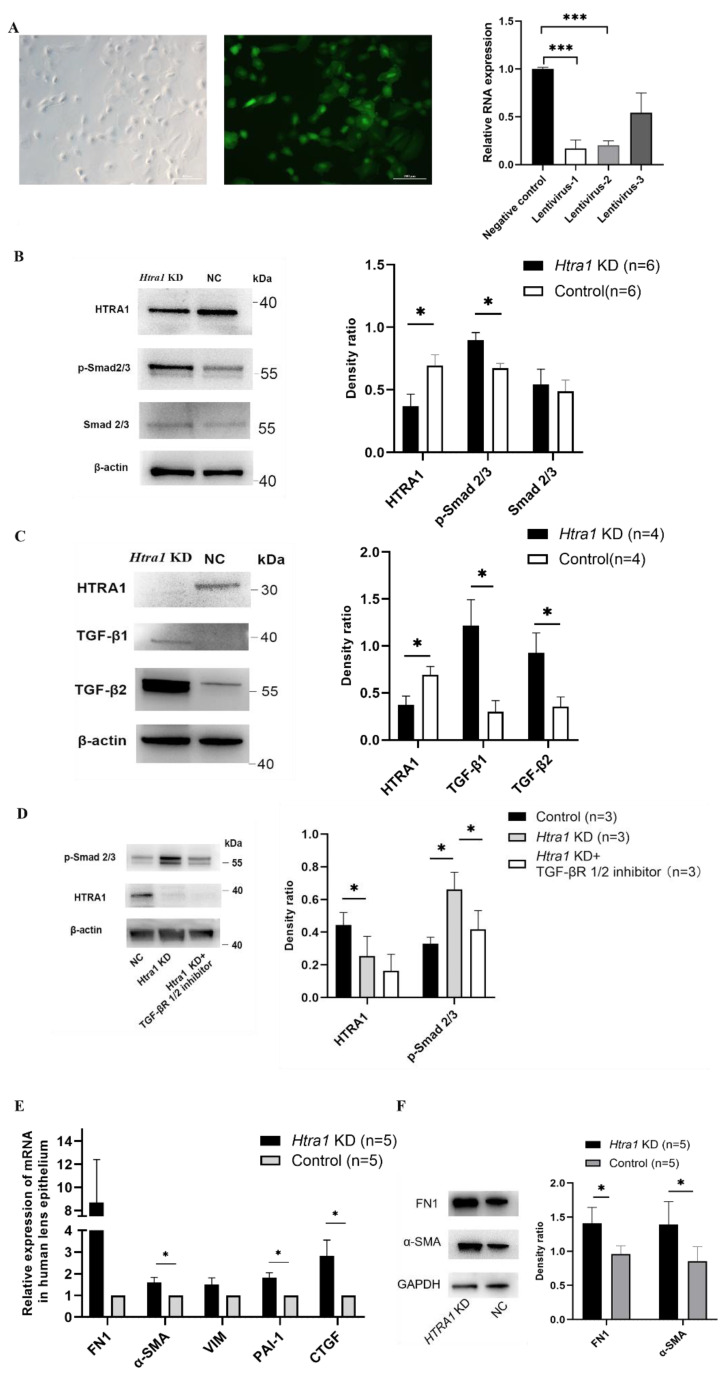
Downregulation of HTRA1 directly activated the TGF-β/Smad signalling in primary HLECs. (**A**) Primary HLECs after LV-HTRA1-RNAi lentivirus transfection, the middle image showed the GFP expression, and the right image showed the mRNA levels of HTRA1 after transfection. (**B**,**C**) Changes of HTRA1, Smad2/3, p-Smad2/3, TGF-β1 and TGF-β2 gene expression detected by Western blotting (Smad2/3 *p* > 0.05, p-Smad2/3 *p* = 0.014, TGF-β1 *p* = 0.021, TGF-β2 *p* = 0.039) in primary HLECs collected from PSC eyes after treatment of HTRA1 knockdown (*Htra1* KD). (**D**) Changes of p-Smad2/3 expression detected by Western blotting with HTRA1 knockdown ± TGF-βR1/2 inhibitor treatment (10 ng/mL for 24 h) in primary HLECs (p-Smad2/3 *p* = 0.042, 0.027, respectively). (**E**) The mRNA levels of α-SMA, PAI-1 and CTGF elevated in HLECs after treatment of HTRA1 knockdown (*p* = 0.022, 0.005, and 0.038, respectively). (**F**) The protein levels of FN-1, and α-SMA in HLECs after treatment of HTRA1 knockdown (*p* = 0.017 and 0.031, respectively). * *p* < 0.05, *** *p* < 0.001.

**Figure 5 ijms-23-14431-f005:**
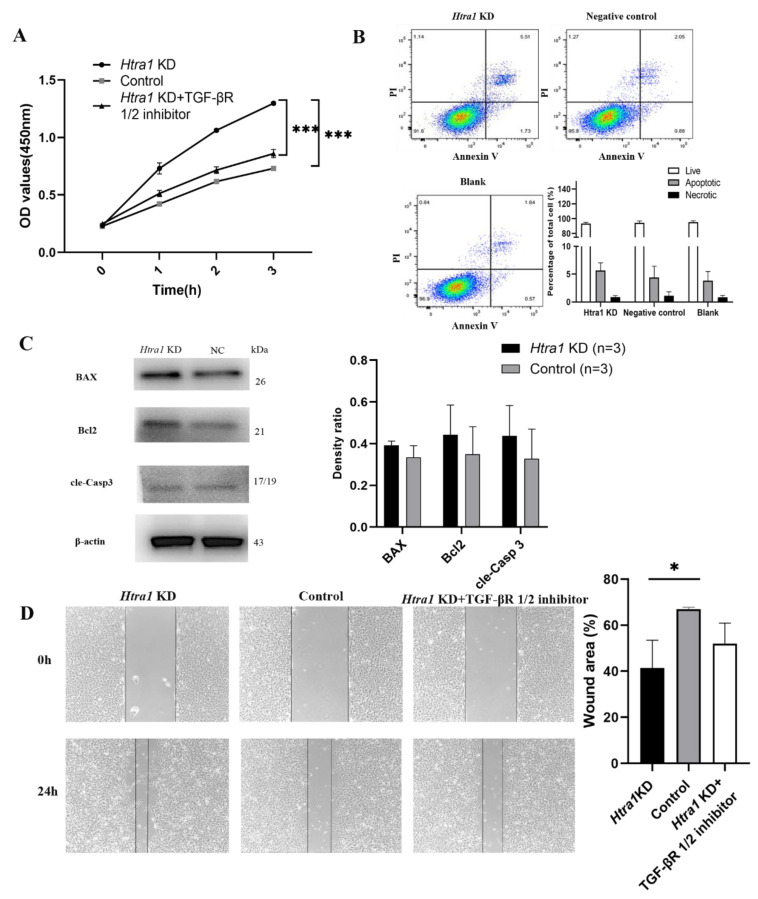
The proliferation, migration and apoptosis changes of HLECs (SRA 01/04) after treatment of HTRA1 knockdown. (**A**) CCK-8 assay showed promoted growth ability of HLECs after downregulated expression of HTRA1 (*p* < 0.001) and was attenuated after treatment of HTRA1 knockdown and the TGF-βR1/2 inhibitor (*p* < 0.001). (**B**) Annexin V/7-AAD staining and flow cytometry did not show significant apoptotic cell changes after treatment of HTRA1 knockdown. (**C**) The protein levels of BAX, Bcl2 and cleaved Caspase-3 in HLECs after treatment of HTRA1 knockdown (*p* > 0.05). (**D**) The scratch healing assay showed knockdown of HTRA1 promoted the migration of HLECs (*p* = 0.022), and was attenuated after treatment of HTRA1 knockdown and the TGF-βR1/2 inhibitor. * *p* < 0.05, *** *p* < 0.001.

**Figure 6 ijms-23-14431-f006:**
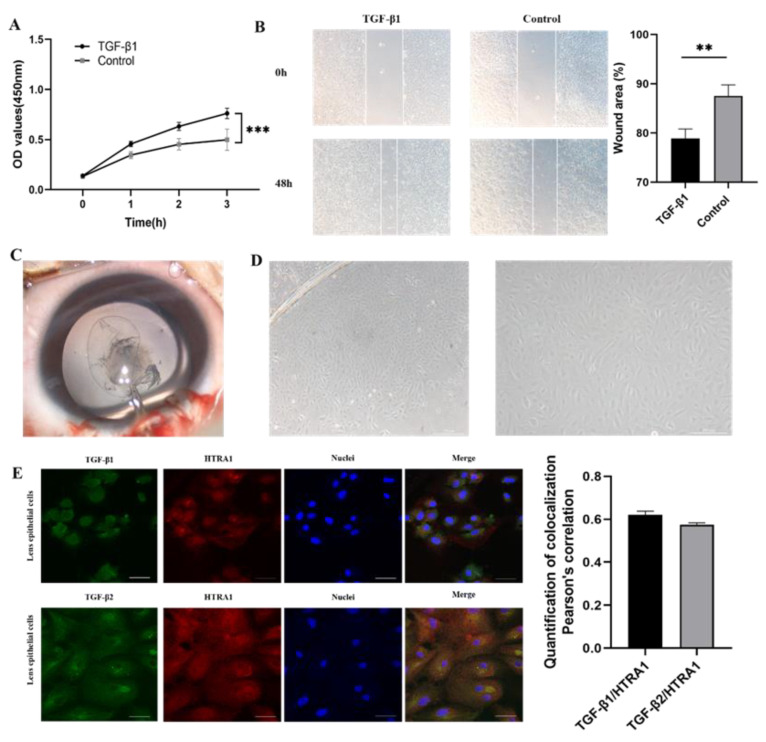
Elevated proliferation and migration of lens epithelial cells by TGF-β1 and the colocalization between HTRA1 and TGF-β1, as well as TGF-β2. (**A**) CCK-8 assay of human lens epithelial cells after TGF-β1 treatment (*p* < 0.001). (**B**) Scratch healing assay of human lens epithelial cells after TGF-β1 treatment (24 h, *p* = 0.007). (**C**) Anterior lens capsules were obtained via capsulorhexis during cataract surgery. (**D**) Cultured primary lens epithelial cells showed migration from the rim of lens capsules. (**E**) Immunofluorescence images of HTRA1, TGF-β1, and TGF-β2 staining. The amount of colocalization was analysed. ** *p* < 0.01, *** *p* < 0.001, Scale bar: 20 μm.

**Figure 7 ijms-23-14431-f007:**
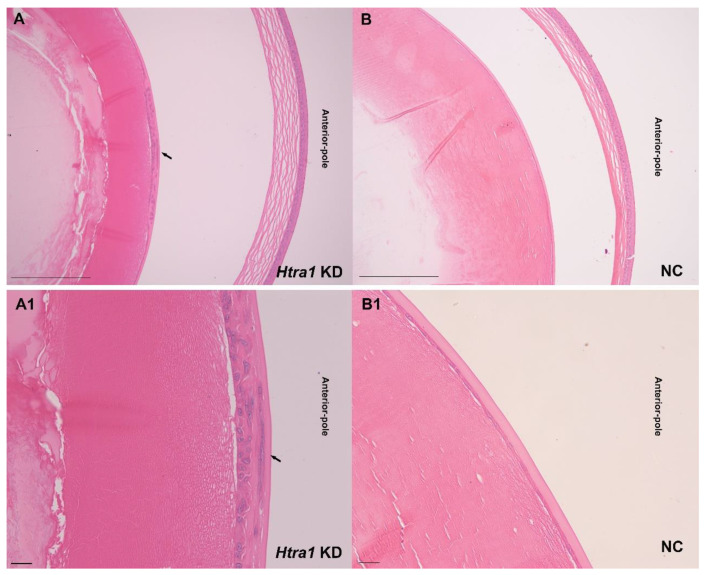
Images of haematoxylin and eosin staining after knocking down HTRA1 by rAAV injection in mouse eyes. Aberrant proliferation and migration of lens epithelial cells were observed in (**A**,**A1**) (Arrow). (**A**,**B**): Scale bar: 200 μm, (**A1**,**B1**): Scale bar: 50 μm. A and A1, as well as B and B1, were images from the same position.

**Figure 8 ijms-23-14431-f008:**
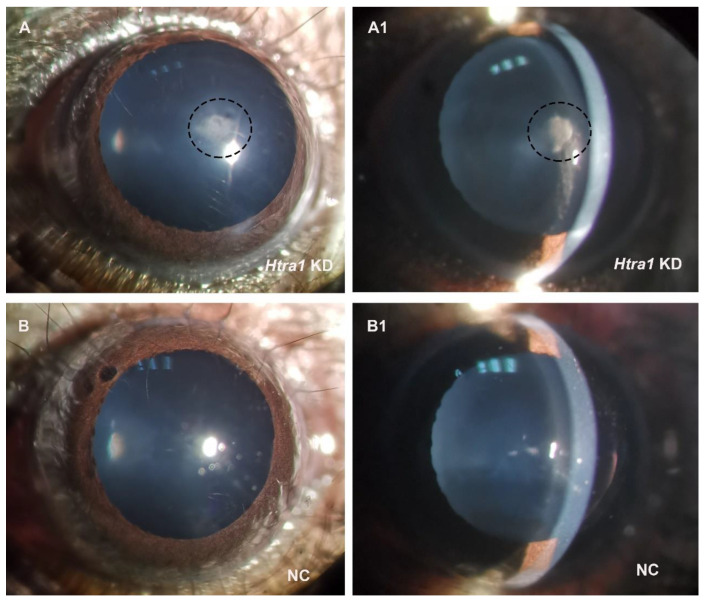
Images of mouse eyes after treatment of HTRA1 knockdown. (**A**,**A1**) showed that downregulation of HTRA1 induced the formation of subcapsular cataract. (**A**,**A1**), as well as (**B**,**B1**), were images from the same position.

**Figure 9 ijms-23-14431-f009:**
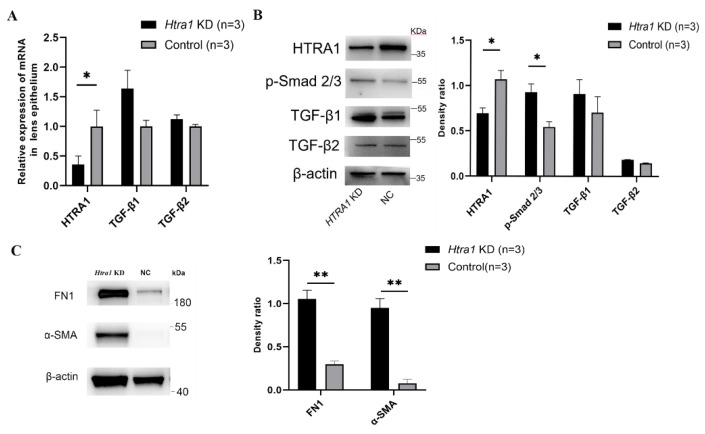
Downregulation of HTRA1 activated the TGF-β/Smad signalling in primary mouse lens epithelial cells. (**A**) Changes of HTRA1, TGF-β1 and TGF-β2 gene expression detected by RT-qPCR (HTRA1 *p* = 0.045) in primary mouse lens epithelial cells after treatment of HTRA1 knockdown. (**B**) Changes of HTRA1, p-Smad2/3, TGF-β1 and TGF-β2 gene expression detected by Western blotting (HTRA1 *p* = 0.029, p-Smad 2/3 *p* = 0.024) in primary mouse lens epithelial cells after treatment of HTRA1 knockdown. (**C**) The protein levels of FN-1 and α-SMA in primary mouse lens epithelial cells after treatment of HTRA1 knockdown (*p* = 0.002 and 0.002, respectively). * *p* < 0.05, ** *p* < 0.01.

## Data Availability

The data that support the findings of this study are available from the corresponding author, Y.L., upon reasonable request.

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
