# Peer review of "TGF-β/Smad Signalling Activation by HTRA1 Regulates the Function of Human Lens Epithelial Cells and Its Mechanism in Posterior Subcapsular Congenital Cataract"

_ijms, 2022, doi:10.3390/ijms232214431_

Round 1
Reviewer 1 Report
• The logic of the research design is clear and presented well. The background was clear, concise, and informative.
Several comments:
l Primary HLECs and HLEC line were used to performed these experiments. Could the authors explain which cells used in each experiment?
l It would be better to use WB or Wes to determine proteins semi-quantitatively in isolation but not combined use of both.
l Figure 3. TGF-βR1, CTGF and PAI-1 were only detected by RT-qPCR analyses. Smad2/3 and p-Smad2/3 were only detected by WB. Why not use both RT-qPCR analyses and WB to detect these proteins? Could the authors explain this result in greater detail?
l Figure 5. No data shows the results of proliferation and migration of primary HLECs after treatment of HTRA1 knockdown and TGF-βR1/2 inhibitor. It would be better to perform the CCK-8 assay and scratch healing assay in this experiment.
l Figure 8. It would be helpful to show the lens epithelial cells proliferation and migration in posterior subcapsular position for the background concerning about PSC, and mark the position of lens epithelial cells(anterior or posterior subcapsular) in figure 8A.
l Figure 9 should also include common HTRA1 target proteins, e.g. FN-1 and α-SMA.
Reviewer 2 Report
I congratulate all the authors for this interesting work. In general, I find your work successful, but below are some of my comments about the article.
1. What is this manuscript different from the same team's previously published article in the Gels as "Transcriptomics Analysis of Lens from Patients with Posterior Subcapsular Congenital Cataract" (DOI: doi.org/10.3390/genes12121904). Please explain. Are these studies complementary? Both studies have the same ethics committee approval number.
2. The authors stated that they received ethics committee approval for their work. However, only one ethics committee number was given. Ethics committee approval has not been obtained for tissue samples collected from children?
